# The Impact of a Novel Immersive Virtual Reality Technology Associated with Serious Games in Parkinson’s Disease Patients on Upper Limb Rehabilitation: A Mixed Methods Intervention Study

**DOI:** 10.3390/s20082168

**Published:** 2020-04-11

**Authors:** Patricia Sánchez-Herrera-Baeza, Roberto Cano-de-la-Cuerda, Edwin Daniel Oña-Simbaña, Domingo Palacios-Ceña, Jorge Pérez-Corrales, Juan Nicolas Cuenca-Zaldivar, Javier Gueita-Rodriguez, Carlos Balaguer-Bernaldo de Quirós, Alberto Jardón-Huete, Alicia Cuesta-Gomez

**Affiliations:** 1Department of Physical Therapy, Occupational Therapy, Rehabilitation and Physical Medicine, Universidad Rey Juan Carlos, PC 28922 Madrid, Spain; patricia.sanchezherrera@urjc.es (P.S.-H.-B.); roberto.cano@urjc.es (R.C.-d.-l.-C.); alicia.cuesta@urjc.es (A.C.-G.); 2Robotics Lab, University Carlos III of Madrid, Leganés, PC 28911 Madrid, Spain; eona@ing.uc3m.es (E.D.O.-S.); balaguer@ing.uc3m.es (C.B.-B.d.Q.); ajardon@ing.uc3m.es (A.J.-H.); 3Department of Physical Therapy, Occupational Therapy, Rehabilitation and Physical Medicine, Research Group of Humanities and Qualitative Research in Health Science of Universidad Rey Juan Carlos (Hum&QRinHS), Universidad Rey Juan Carlos, PC 28922, Madrid, Spain; jorge.perez@urjc.es (J.P.-C.); javier.gueita@urjc.es (J.G.-R.); 4Rehabilitation Unit, Hospital de Guadarrama, Department of Physical Therapy, Universidad Francisco de Vitoria, PC 28223 Madrid, Spain; nicolas.cuenca@salud.madrid.org

**Keywords:** Parkinson’s disease, biomedical technology, biomedical enhancement, neurological rehabilitation, mixed methods research

## Abstract

Background: Parkinson’s disease is a neurodegenerative disorder that causes impaired motor functions. Virtual reality technology may be recommended to optimize motor learning in a safe environment. The objective of this paper was to evaluate the effects of a novel immersive virtual reality technology used for serious games (Oculus Rift 2 plus leap motion controller—OR2-LMC) for upper limb outcomes (muscle strength, coordination, speed of movements, fine and gross dexterity). Another objective was to obtain qualitative data for participants’ experiences related to the intervention. Methods: A mixed methods intervention (embedded) study was used, with a qualitative design after a technology intervention (quantitative design). The intervention and qualitative design followed international guidelines and were integrated into the method and reporting subheadings. Results: Significant improvements were observed in strength (*p* = 0.028), fine (*p* = 0.026 to 0.028) and gross coordination dexterity, and speed movements (*p* = 0.039) in the affected side, with excellent compliance (100%) and a high level of satisfaction (3.66 ± 0.18 points out of the maximum of 4). No adverse side effects were observed. Qualitative findings described patients’ perspectives regarding OR2-LMC treatment, facilitators and barriers for adherence, OR2-LMC applications, and treatment improvements. Conclusions: The intervention showed positive results for the upper limbs, with elements of discordance, expansion, and confirmation between qualitative and quantitative results.

## 1. Introduction

Parkinson’s disease (PD) is a neurodegenerative disorder that predominately affects dopamine-producing neurons in the substantia nigra [1]. Typical PD motor symptoms include resting tremor, which mainly occurs at rest and is described as a pill-rolling tremor in the hands. However, other forms of tremor are possible, such as bradykinesia, rigidity, and gait and balance problems. PD also has a large number of non-motor symptoms [2]. Dexterity, gross, and fine motricity impairments are among the most disturbing symptoms impacting activities of daily living, even in mild to moderate stages of the disease [3].

Multidisciplinary input is increasingly important in the management of PD [4]. Rehabilitation treatment, alongside the well-established pharmacological and surgical interventions, is now encouraged as an adjunctive treatment starting from the early stages of PD [5,6]. In this context, the use of computer-based virtual reality (VR) technology is a promising new rehabilitation tool with a wide range of potential applications [7]. It allows users to interact with simulated environments and receive feedback on their performance in real-time scenarios through task-oriented training. VR technology can also optimize motor learning in a safe environment. It may serve as a worthy alternative to conventional approaches of PD physiotherapy treatment [5] by helping users to learn new motor strategies and relearn lost motor abilities. Additionally, VR technology can engage patients in long-term exercise programs, providing a challenging and motivating training environment while replicating real-life scenarios that facilitate the transition to the functional activities of daily living [8,9]. However, it remains unclear exactly how immersive VR technology can be optimally used in PD patients with impaired upper limb (UL) dexterity. 

Experimental interventions utilizing new technologies for disease treatment and rehabilitation should be analyzed for efficacy and safety, as well as patient satisfaction, perspective, and experience [10]. Therefore, it is necessary to study both quantitative (e.g., the effectiveness of the intervention) and qualitative (patient acceptance or rejection of the therapy) aspects regarding the application of new technology [11]. Mixed methods research (MMR) is an approach that combines the strengths of quantitative and qualitative research to obtain a richer and deeper understanding [12]. MMR has been used previously in research involving health services, such as program evaluation, community health, the implementation of innovative interventions [12], clinical issues, health care organizational performance, and health care decision making, including a supplemental qualitative component within experimental or quasi-experimental studies of complex interventions [11]. Previous studies have described the use of MMR in PD patients by combining the analysis of the effects of experimental interventions with qualitative investigations into the impact of various treatments [13,14].

The Qualitative Research in Trials (QUART) study [10] reported how qualitative research of health technologies had been used in trials and identified ways to maximize the value of a trial to provide evidence of a treatment’s effectiveness. QUART provided the pathway to use MMR for qualitative analysis of a new technological intervention in efficacy studies (quantitative component). This approach determines whether an intervention is delivered as intended, describes implementation processes, generates an understanding of why the intervention either worked or failed, and demonstrates whether the effectiveness of therapy is promoted or limited in real-world situations. Qualitative analyses can occur before, during, or after the quantitative study of the intervention [15]. Before applying treatments based on novel technologies to larger sample sizes, these innovations must be progressively tested to describe their efficacy, safety, and patient acceptance [10]. Similarly, new treatments must also be investigated under real clinical conditions, where patients and their environment (outside of the hospital) do not necessarily adhere to ideal experimental criteria [16,17]. Such issues are common in complex patients, such as those with PD, who usually suffer from several comorbidities, take various medications, and have different levels of functional impairment [18].

Therefore, the current study aimed to utilize a concurrent embedded design to investigate the use of a novel VR technology in patients with PD. The study included both quantitative and qualitative objectives. The objective of the quantitative design had two parts: to implement a novel immersive VR technology with an Oculus Rift2 plus a leap motion controller (OR2-LMC), using serious games specifically designed by the research team for PD patients; and to analyze its effects on muscle strength, coordination, speed of movements, and fine and gross dexterity. The qualitative design objective, which was created to assess satisfaction and compliance levels, was used to explore and describe the subjects’ experiences with and understanding of the OR2-LMC intervention during upper limb rehabilitation, as well as to describe the polarity (acceptance or rejection) of this novel technological treatment.

## 2. Materials and Methods

We used a mixed methods intervention (MMI) study design with a qualitative component, a concurrent embedded design [15]; quantitative methodology was used in the first step of data analysis, while the second step used qualitative methodology. Qualitative data was collected after the OR2-LMC intervention to elucidate potential mechanisms and explain the quantitative outcomes (participant experience, improvements related to the intervention, and potential mediating and moderating factors) [15]. Table 1 summarizes the MMI used in this study. The quantitative and qualitative data were integrated at the method level through embedding one within the other [15], and at the interpretation and reporting level through narrative and joint displays [19]. Figure 1 details the mixed methods design and embedded integration that were used.

Findings from the quantitative and qualitative methodologies were integrated to understand the relationships between the quantitative and qualitative components (the integration phase of the study) [15]. The objective of this phase is to balance the respective strengths and weaknesses of various methods to maximize the yield of distinct, potentially complementary sources of evidence [15].

In this study, we followed the Best Practices for Mixed Methods Research in the Health Sciences of NIH [20] and the guidelines provided in Good Reporting of a Mixed Methods Study [21]. Additionally, the quantitative and qualitative design components of the study followed appropriate quality guidelines. The quantitative intervention phase used the template for intervention description and replication (TIDieR) checklist [22,23], which provides a standardized template for the description of all elements necessary for reports on a non-pharmacological intervention [23] (see Appendix A).

We followed the guidelines for qualitative studies established by the Consolidated Criteria for Reporting Qualitative Research [24] and the Standards for Reporting Qualitative Research [25]. Additionally, we followed the criteria for guaranteeing the trustworthiness of qualitative research, as proposed by Guba and Lincoln [26,27]. The various techniques performed and the application procedures used to control trustworthiness are described in Appendix A. 

### 2.1. Quantitative Intervention Phase Design

In the first quantitative research phase, we conducted a non-randomized clinical intervention involving PD patients, as detailed below.

#### 2.1.1. Sampling and Participants

Participants were recruited from the PD Patients Association using a no-probabilistic sampling method for non-consecutive cases. Inclusion criteria were: a PD diagnosis following the Brain Bank of the United Kingdom [28], severity between I–IV stage per the Hoehn and Yahr scale [29], participants with >60% (some dependency; can do most chores, but exceedingly slowly and with much effort; errors; some chores impossible) on the Schwab and England scale [30], and subjects with a stable or slightly fluctuating motor response to pharmacological treatment.

Exclusion criteria were: patients receiving specific UL rehabilitation treatment at the time of the study, off phase of pharmacological treatment, ≤24 points on the Mini-Mental Status Examination [31], injuries affecting the UL or presence of diseases other than PD, I and V stages of the Hoehn and Yahr scale, visual impairments not correctable by glasses, and those unable to sign the informed consent.

#### 2.1.2. Intervention

Previous studies [2] demonstrated the efficacy of serious games based on leap motion for improving the functionality of the UL in patients with PD. The TIDieR checklist [22,23] was followed (Appendix A). Patients used LMC systems mounted on an OR2 device with their elbows positioned at an initial flexion of 90° while seated at a table at mid-trunk height. The video games developed for this study were designed to empower depth perception to maximize the functional gains from the therapy. Four video games were developed using the technical specifications and guidelines provided by the clinicians. The development of each video game emphasized training-specific UL functionalities, such as reaching, grasping, pronation, or different combinations of these. Some cognitive training was also included in the video games using memory exercises.

Treatments were conducted at the APARKAM Association (Asociación de Parkinson de Alcorcon y Municipios) between January and March of 2019. Individual sessions lasted for 30 min and were conducted three times per week over six weeks, for a total of 18 sessions per patient. Patients performed the four video games in the following order (Figure 2): the reach game (RG), the sequence game (SG), the grab game (GG), and then the flip game (FG). The duration of each game depended upon the individual skill level of the patient, with the average duration of each game lasting approximately five to seven minutes.

#### 2.1.3. Data Collection and Outcome Measures

A Jamar Hydraulic Hand Dynamometer was used. This hand dynamometer is an easy-to-use method for routine screening of grip strength and hand functionality, and can measure the isometric force (in pounds and kilograms). It is also one of the most used objective tools for grip strength assessment, with excellent reliability and sensitivity [32,33]. Each patient performed three grip movements for each side (more affected and less affected). Mean values for each side (more and less affected) were registered.

The box and block test (BBT) measures unilateral gross manual dexterity and can be used for different tests, including patients with PD. The BBT is a wooden box divided into two compartments and 150 blocks. The test consists of moving the maximum number of blocks one-by-one from one compartment of a box to another of an equal size within 60 s. The test begins with the unaffected UL to register scores and both sides are tested. The BBT has shown its validity in subjects with UL disability [34,35].

The Purdue pegboard test (PPT) is a test used to measure coordination and speed of movement of the hands, fingers, and arms, along with assessment of fingertip dexterity. It has been validated in patients with impairments of the ULs. The testing board consists of a board with 4 cups across the top and two vertical rows of 25 small holes down the center. The two outside cups contain 25 pins each; the cup to the immediate left contains 40 washers, and the cup to the immediate right of the center contains 20 collars. The following five subtests were conducted: (a) right hand (30 s): patients used their right hand to place as many pins as possible down the row within 30 s; (b) left hand (30 s): patients used their left hand to place as many pins as possible down the row within 30 s; (c) both hands (30 s): patients used both hands simultaneously to place as many pins as possible down both rows; (d) assembly (60 s): patients used both hands simultaneously while assembling pins, washers, and collars. Three trials for each subtask were conducted, and the mean score was registered. The PPT is a reliable method for dexterity assessment in PD patients [36].

The action research arm test (ARAT) is a 19-item measure divided into four subtests (grasp, grip, pinch, and gross arm movement). Each item is rated on a 4-point scale as follows: (0) can perform no part of the test; (1) performs the test partially; (2) completes the test, but takes an abnormally long time or has great difficulty; (3) performs the test normally. The maximum score for the test is 57 points. This assessment has been validated for PD patients [37,38].

The client satisfaction questionnaire (CSQ-8) was developed to assess global client satisfaction through eight questions (quality of service, type of service, needs met, recommend to a friend, amount of help, deal with problems, overall satisfaction, and come back). Answers are coded using a 4-point Likert scale (from 1 to 4). Total possible scores range from 8 to 32, with higher scores indicating greater satisfaction [39,40]. Additionally, we recorded the attendance rate (%) for therapy sessions (compliance) and possible side effects related to the intervention. 

All outcome measures were conducted by three expert raters trained in all assessments and blinded to the interventions (rater 1 conducted Jamar and BBT assessments; rater 2 conducted PPT measures; rater 3 conducted ARAT and CSQ-8 evaluations). Three evaluation periods were determined: before any intervention, in a post-treatment period (after six weeks of treatment), and a follow-up session (one month without rehabilitation treatment).

#### 2.1.4. Data Analysis

The SPSS statistical software system (SPSS Inc., Chicago, IL, USA; version 24.0) was used for statistical analysis. The Shapiro-Wilk’s test and the Kolmogorov-Smirnov test were employed to verify the normality of data distribution. Furthermore, the Wilcoxon test for related samples was used to compare variables. The statistical analysis was calculated with a 95% confidence level. A *p*-value < 0.05 was considered significant. The mean and the standard deviation were used to calculate the effect size for the comparisons using Cohen’s d statistic. Mean differences of 0.2, 0.5, and 0.8 standard deviations are considered small, medium, and large effect sizes, respectively.

### 2.2. Qualitative Phase Design

A qualitative exploratory case study was conducted in our study [26,41].

#### 2.2.1. Sampling and Participants

Purposeful sampling methods were employed based on their relevance to the research question and were not based on clinical representativeness [42]. All patients were recruited from the OR2-LMC trial. The recruitment took place when the patients finalized their intervention at the APARKAM Association. If patients met the inclusion criteria for our study and agreed to participate, they were included in the qualitative phase. The sampling process was based on the information power criteria established by Malterud et al. [43]. Information power indicates that the more information the sample holds that is relevant for the study, the lower the number of participants that is needed. For this reason, we included the same participants that were recruited for the intervention in the quantitative phase. No participants withdrew from the study.

#### 2.2.2. Data Collection

Semistructured in-depth interviews were the main tool used for data collection. Interviews were based on a question guide designed to gather information about specific topics of interest [26] (Appendix A). The question guide was developed based on a prior literature review [44,45] and the researchers’ experience [26]. All the interviews were recorded and transcribed verbatim, with 311 min of interviews overall. The interviews were held at the patient’s home or the PD Patients Association, depending on the patient’s preference. We also collected the researchers’ field notes, which provided a rich source of information and supported the interview data [26]. No third party was present at the interviews. All interviews were in Spanish.

#### 2.2.3. Data Analysis

Literal transcriptions were made from each interview and researchers’ field notes [26,42]. A thematic analysis began by analysis of the most descriptive content to arrive at meaningful units and then went into further depth to produce thematic code groups [26,46]. Clusters of meaningful units were generated, combining meaningful units on the same issue or with the same content, until the main topics (themes) emerged [26,46]. This codification procedure was conducted separately for the interviews and the field notes. Additionally, a matrix was built with the results obtained from the analysis. Subsequently, joint meetings of the research team were held to combine the results, and a consensus was established for differences in theme identification. Subsequently, the research team held joint meetings to show, combine, integrate, and identify final themes. Additionally, the Bing Sentiment Dictionary [47] and the SODictionariesV1.11Spa2 [48] were used to analyze the content of the free text and obtain a description of the acceptance or rejection (polarity) [49] of the OR2-LMC treatment. Content analysis has been used previously in the health sciences to study medical and patient narratives [50,51]. Four phases were used progressively for the analysis of acceptance-rejection (polarity). First, we created a file with the text of the interviews, which was broken down by phrases for textual analysis. Second, we calculated the polarity using the Bing Sentiment Dictionary [47], the amplifiers and de-amplifiers from SODictionariesV1.11Spa2 [48], and the negators proposed by Vilares et al. [49] (Appendix A). Third, we calculated the scatterplot of the sentences in the text regarding neutrality to identify positive or negative trends. Finally, the evolution of the emotional valence (positive-negative) was shown throughout the interviews. We applied a Fourier transformation to confirm the polarity trend.

### 2.3. Ethical Considerations of the Mixed Methods Intervention

The study was approved by the Clinical Research Ethics Committee of the University Rey Juan Carlos (Project number: 2903201707617). Furthermore, we followed the principles articulated in the WMA Declaration of Helsinki [52]. Each patient signed the informed consent and permission to record the interviews before the quantitative and qualitative studies.

### 2.4. Embedded Integration Procedure for Quantitative and Qualitative Content

How qualitative and quantitative data fit together is a critical concept in developing a mixed methods study; this issue remains relevant throughout the entire process [15,19]. In the current study, quantitative and qualitative data were integrated at the method level through embedding one within the other [53], and at the interpretation and reporting levels through narrative and joint displays [15,19] (Figure 1). Embedding at the method level occurs in studies with both primary and secondary questions (objectives) when different methods are employed to address each question. We implemented data integration by (a) analyzing the primary dataset to answer the primary research question (quantitative intervention); (b) analyzing the secondary dataset (qualitative) embedded within the primary design and incorporating the secondary results; (c) interpreting how the primary (quantitative) and secondary (qualitative) results answered the quantitative and qualitative questions; and (d) presenting the complete set of findings [15]. Also, the narrative approach (contiguous) and joint displays (figures and graphs) were used to interpret and present the findings [15,19]. A contiguous approach to integration involves the presentation of findings within a single report, although the quantitative and qualitative findings are reported in different sections. Integration through joint displays brings the data into a visual medium that enables one to draw new insights beyond what can be gained through the results of the separate quantitative and qualitative methods. For these reasons, we organized the data into figures, tables, and graphs [19,54].

## 3. Results

We report our results in the following order: (1) quantitative and intervention results, (2) qualitative results, and (3) mixed methods findings (integration) [53].

### 3.1. Quantitative Findings

Of the eight patients selected at the onset of the study, our final sample consisted of six patients (P1–P6), with five males and one female. Two subjects were excluded due to their inability to attend the assessment or treatment sessions. The ages of the patients ranged from 69 to 80 years (mean age 74.50 ± 4.72 years). The left UL was the side most affected by PD in four of these patients, while the right UL was the most affected in the remaining two patients. The Schwab and England activities of daily living scores of the patients ranged from 60% to 100% independence (71.66 ± 9.83%) (Table 2). 

Statistical analysis demonstrated significant improvements in grip strength in both ULs post-treatment (*p* = 0.028 for the more and less affected sides), as well as in the less affected UL during the pre-follow-up measurement (*p* = 0.028). The effect size was medium (>0.50) for the Jamar test pre-post assessment for the more and less affected side, and the effect size was small (>0.20) for the Jamar test in the follow-up assessment for the less affected side. Significant improvements were also found in the BBT for the more affected UL pre-post assessment (*p* = 0.039), with a small (>0.20) effect size. Significant improvements were observed for the PPT for the more affected (*p* = 0.027) and less affected UL (*p* = 0.028) pre-post assessment, with a medium (>0.50) effect size. The PPT using both ULs during the pre-follow-up assessment (*p* = 0.026) and the PPT assembly during the pre-post assessment (*p* = 0.028) with a small (>0.20) effect size (Table 3). 

Patients showed a high degree of satisfaction as measured with the CSQ-8. Results showed a mean of 3.66 (±0.18) points out of the maximum of 4. Of the eight items in this questionnaire, all participants gave the highest score to question 5 (“Are you satisfied with the help you have received?”) and question 7 (“In general, are you satisfied with the services you have received?”). None of the participants expressed disagreement or dissatisfaction in the remaining questions (Table 4). Furthermore, the attendance rate was 100% for therapy sessions, and no adverse side effects were observed in the intervention.

### 3.2. Qualitative Findings

#### 3.2.1. Results of the Thematic Analysis

We extracted the themes that represented the participants’ experiences through analysis of the collected qualitative data, which included interviews, researchers’ notes, and personal letters. Four themes emerged: (1) patients’ perspectives regarding the OR2-LMC treatment, (2) facilitators and barriers related to OR2-LMC treatment adherence, (3) management and application of the OR2-LMC treatment, and (4) potential improvements to the treatment. 

Patients’ Perspectives Regarding OR2-LMC Treatment:

All patients reported that the OR2-LMC treatment was not better at increasing their upper limb functionality than the conventional treatment (physiotherapy, occupational therapy, etc.). Patients showed that this new treatment did not replace conventional treatment, but rather complemented it. Some patients (P1, P2) argued that health professionals reinforced the importance of movement-oriented treatments and the avoidance of immobility. For some patients (P2, P4–P6), the new treatment was more of a mental challenge than a physical one. None of the patients spontaneously reported any improvements following OR2-LMC treatment. However, some did perceive improvements in their daily activities, such as eating, handling utensils, buying food, and checking tickets at home (P2–P4, P6). Additionally, some patients perceived improvements post-treatment when driving, looking at traffic lights on the street, and reading. Some patients also felt as though they improved coordination, joint movement, concentration, mental speed during activities, and the ability to overcome obstacles on the street.

Facilitators and Barriers Contributing to OR2-LMC Treatment Adherence:

Patients were asked to identify barriers and facilitators that contributed to their adherence to the OR2-LMC treatment.

Facilitators: (a) A sense of competition against the machine was perceived as a facilitator by all patients. Improvement of personal scores in the video games, as well as winning them, was seen as a stimulus to continue with therapy; winning was perceived as something that depended on their upper limb abilities. (b) One patient (P2) reported that overcoming these challenges made him feel closer to his family and more competent in his daily living activities. (c) A sense of frustration was felt when failing to overcome a challenge (P2–P6) and the treatment helped patients to identify their limits while striving to overcome them (P1–P6), which gave a greater sense of satisfaction (P4–P6). (d) The OR2-LMC treatment helped patients become more aware of situations that they had previously paid little attention to, such as picking up and manipulating small objects (P1, P4–P6). (e) The games helped patients to focus on their treatment and be more involved in it. (f) The treatment encouraged some patients (P2–P6) to compare their scores with each other and share their experiences with the therapy by discussing the challenges they overcame, supporting each other, and feeling that they were all facing these new games together.

Barriers: (a) Fatigue (P1, P6) from a sense of tension and nervousness from wanting to do their best in the video games (P1). (b) The short time interval available to become acquainted with the virtual world and perform the required activities (P1, P2, P5, P6). Some patients felt that their low scores did not accurately reflect their actual health status, and instead demonstrated their lack of prowess with VR (P1, P2, P5). (c) The monotony of the video game activities (P2–P4). (d) Fear of new challenges and activities involved in the therapy (P6), and the sense of frustration in failing to overcome a challenge (P4, P6). (f) The PD tremors interfered with their ability to perform the tasks (P4, P5).

Management and Application of the OR2-LMC Treatment:

There was a process of adaptation to the new treatment (P1–P6). At first, patients reported feeling awkward as they tried to adapt to the “virtual world”, although they did gradually become more comfortable and improved their performance. The patients reported that the treatment seemed inapplicable to the home setting (P1, P2, P4, P5), because the system requires a lot of physical space, is complex to assemble, demands previous knowledge and skills for use, and requires a lot of time and money. The patients also described feeling that the treatment should be administered by a qualified professional (P1, P2, P4–P6), in order to prepare and operate the equipment and resolve any unforeseen events that may occur. Moreover, one patient (P1) felt that a professional was required to monitor and track the results of the treatment. In contrast, another patient (P2) felt that a professional was needed to correct his actions and help guide him through the correct performance.

Regarding potential help from their families, some patients (P3–P6) preferred to be monitored by a professional, although they felt their families could help them if necessary. The reasons given for preferring professional help included a feeling of safety, as well as the inability to postpone treatment or falsify their results (P5). One patient (P2) justified the feeling that family members should not be responsible for applying the treatment with the following statement: “families already deal with enough seeing how we deteriorate a little more every day.”

Potential OR2-LMC Treatment Improvements:

The patients described improvements they felt could be included for newer versions of the treatment, including: (a) competition among the users, because the competition was perceived as a positive element of the treatment (P1); (b) conducting preparatory sessions to increase familiarity with VR before treatment (P1, P2); (c) expanding the catalogue of games and activities available to increase motivation (P2, P3) and help with the treatment of additional symptoms of PD, such as tremor (P2) and lack of balance (P3); (d) clearly explaining the treatment, its application, and realistic expectations of results during recruitment (P1, P2, P4, P5), because some patients either did not fully understand what the treatment involved (P2) or had unrealistic expectations of potential improvements (P3–P5); (e) including various levels of difficulty in the video games to stimulate continuous efforts (P2, P3); and (f) requiring treatment administrators to experience the treatment before applying it to gain the first-hand experience with the technology (P3).

#### 3.2.2. Results of the Acceptance-Rejection (Polarity) Analysis

The analysis showed that the overall polarity of the interviews was neutral, with a slight tendency towards positivity. The polarity scatter plot (Figure 3) showed that the phrases (n = 1000) extracted from the patient interviews were grouped around neutrality (0.068 ± 0.301). The regression line of this plot appeared stable, and the phrases were grouped symmetrically within both polarities (acceptance-rejection). However, extreme values appeared at positive valence levels greater than one, causing the global deviation to shift towards positivity. Therefore, the scatter plot confirmed that the overall polarity of the interviews was neutral, with a slight tendency towards positivity (acceptance).

Examining the evolution of the patients’ emotional valences (positive-negative) during the interviews demonstrated slightly positive neutrality that remained stable throughout the interview (Figure 4).

The percentage of emotions per valence appeared stable, with two positive moments at 50% and 90% of the interview text (Appendix A). After that, the polarity remained stable around neutrality according to the Fourier transformation, although the trend was more positive in the initial 50% of the discourse and more negative in 60–80% of the narrative path (Appendix A). Finally, analyzing the six interviews as a whole, a neutral and slightly positive polarity (acceptance) was observed. However, the last two interviews (P5, P6) were markedly more negative (rejection) than the others (P1–P4) (Figure 5).

### 3.3. Mixed Methods Findings (Integration)

The integration showed elements of confirmation, expansion, and discordance [19] (Table 5). Our confirmation results demonstrated that the OR2-LMC treatment led to significant improvements in unilateral gross manual dexterity coordination (BBT), speed of movement and fine motor dexterity (PPT), and the CSQ-8. These results were confirmed through the patients’ narratives, which described improvements in patients’ daily living activities post-treatment. Additionally, analyzing the content of patient interviews demonstrated a polarity of acceptance (positive) to the new treatment. However, elements leading to a better knowledge of the phenomenon (intervention effect) were detected due to the questions regarding treatment barriers, facilitators, applications, and improvements. Importantly, there was a disagreement in the quantitative results from the CSQ-8, with very positive results regarding the service during treatment yet with difficulties in understanding the treatment protocols and objectives, with feelings that their expectations of potential benefits from treatment were false or overestimated. Moreover, patients believed the treatment targeted improvement of their mental and cognitive state, rather than the investigated variables of strength, dexterity, and manual coordination.

## 4. Discussion

The main quantitative objective of this project was to implement a novel immersive VR technology through OR2-LMC using specific games designed by the research team. Another objective was to monitor the effects of this treatment in patients with PD concerning their muscle strength, coordination, speed of movements, and both fine and gross dexterity. Statistically, significant improvements were seen in the patients’ grip strength in both their more and less affected ULs post-treatment, as well as in their less affected UL. Statistically significant improvements were also observed in BBT scores when patients used their more affected UL post-treatment, in PPT scores using their more and less affected UL post-treatment, in PPT scores using both hands at the follow-up, and in PPT assembly scores post-treatment with a small to medium effect size.

Few studies have been conducted concerning the use of technology to improve dexterity and coordination in UL rehabilitation [3,55,56,57]. 

Chen et al. [55] developed an immersive VR scenario to treat balance disorders in patients with PD, using visual cues for catching virtual balls in standing and one-step forward positions. Arms and trunk movements were used to assess postural control. Unlike this study, we aimed to design virtual environments to treat UL in PD patients in terms of muscle strength, coordination, speed of movements, and fine and gross dexterity. Daily living activities (coordinated movements of the arms, hands, and fingertips, as well as grasp control) are usually affected in persons with PD. 

In this line, Butt et al. [56] assessed motor symptoms (bradykinesia, frequency, speed and amplitude disturbances, and tremor) in PD patients using an LMC. Sixteen PD patients and 12 healthy subjects performed pronation and supination of the forearms, opening and closing of hands, thumb-forefinger tapping, and postural tremor using the LMC. A neurologist expert in movement disorders explored the same movements according to part III (motor section) of the Unified Parkinson’s Disease Rating Scale (UPDRS). Their results did not show a relationship between the clinical score and the parameters extracted by the LMC, so this system would not be appropriate for assessing UL motor dysfunctions in PD patients. Therefore, we used validated outcome measures for PD patients that are used in clinical settings. The observed improvements for grip strength, BBT, and PPT scores may be due to the designed games that aimed to imitate exercises and movements commonly included in real tasks, such as palmar prehension, finger flexion and extension, or hand pronation-supination. Additionally, some cognitive aspects were included in the training, such as tasks requiring attention and memory. 

Cikajlo and Potisk [57] designed a randomized parallel study with 20 PD patients randomized into two groups—one using 3D Oculus Rift technology plus LMC and the other using a laptop plus LMC. In their study, both groups conducted a pick-and-place task in the virtual world that required precise hand movements to manipulate virtual cubes. The results from their study demonstrated that the use of immersive 3D technology might increase interest and enjoyment in therapy compared to intervention with LMC, and produce faster and more efficient functional performance and reduced resting tremors as assessed by BBT, UPDRS III, and LMC. Our results are in line with Cikajlo and Potisk [57]. Fernández-González [3] conducted a randomized controlled trial using LMC and serious games compared to conventional UL rehabilitation in PD patients. Improvements of coordination, speed of movements, and fine UL dexterity were shown for the experimental group, but no immersive interventions were explored. Therefore, to our knowledge, ours is the first study to evaluate grip strength, coordination, speed of movement, fine and gross dexterity, patient satisfaction, and compliance following the use of the immersive OR2 plus LMC system with serious games designed for PD patients undergoing UL rehabilitation.

Our experimental protocol consisted of 30-min sessions performed three times per week across six weeks (18 sessions), compared to Cikajlo and Potisk [57], who used ten sessions during three weeks of training with 30 min per session. In our research, we followed motor learning principles that encourage the repetition of functional motor tasks in a distributed practice (same treatment time but incorporating resting periods) as a part of a guided learning program [58]. Therefore, the information gained in our study could be valuable in efforts taken to incorporate VR technologies as complementary tools in PD rehabilitation. 

Patient satisfaction with the technology, as measured by the CSQ8 scale, was high. Lack of motivation is a common problem in long-term rehabilitation and leads to reduced treatment adherence. Training in VR can provide a tailored environment with the opportunity to solve motor problems competitively, thus potentially enhancing the motivation to perform repetitive tasks. This motivation, along with more repetitions of arm movement elicited by active gaming compared to traditional rehabilitation [59], may justify the use of VR technology as an adjunct to conventional rehabilitation for PD patients. This potential is especially true when considering the excellent patient satisfaction, increased enjoyment and motivation, and absence of side-effects seen in our study. 

Our qualitative results showed good acceptance of the new treatment. Consistent with previous studies, we identified several aspects of treatment concerning patient effort, interest, feelings of pressure, and acquisition of ability in the virtual world. Cikajlo and Potisk [57] reported that PD patients receiving VR therapy with the Oculus Rift system had a higher level of interest in achieving good results, while also experiencing higher pressure to do well. Importantly, patients’ perceived competency and interest or enjoyment were higher when less pressure or tension was felt, as was seen between the first and last treatment sessions in their study. 

Additionally, our results indicated that our MMR could identify new dimensions of the user’s satisfaction, which a single questionnaire could not. The authors believe that the discordance is a strength in our study because our findings may enrich and expand dimensions evaluated by satisfaction questionnaires of users with chronic motor and cognitive signs and symptoms that relate to a novel immersive VR technology (OR2-LMC). We identified a set of dimensions that would improve the novel immersive technology: the facilitators and barriers for a new treatment, the application of technology treatment by professionals, and dimensions that require improvement. These dimensions are similar to those reported by Tsai et al. [60], who also identified underlying dimensions related to acceptance of e-health technology that complemented CSQ-8. It should be emphasized that these observations do not underestimate CSQ-8, a validated tool that has been applied in web-based interventions (coaching, videoconferencing, and telerehabilitation) for several conditions (e.g., diabetes, chronic obstructive pulmonary disease, stroke, depression, and spinal surgery) and to evaluate preoperative patient satisfaction [60,61,62,63,64].

The current study has several limitations. First, the sample size was small. However, several statistically significant variables showed a medium effect size. Future studies should be conducted with more subjects. Our results cannot be generalized to all PD patients, so these findings should be interpreted with caution. Our findings are related to the ON phase of pharmacological treatment. Moreover, the sampling methods may have resulted in selection bias because patients were recruited from different PD associations. Although our results are encouraging, they are based on a limited number of participants; however, we do not consider this a significant limitation because we used a mixed methods (quantitative and qualitative) approach, with multiple strategies for data collection and analysis, to increase the trustworthiness and credibility of the findings [10,11,15,19,20,21]. Currently, the integration of qualitative and quantitative designs is recommended when evaluating novel technology for use in health sciences [10,11,15]. Further randomized controlled trials will be required to compare our experimental protocol with other conventional approaches to UL rehabilitation and to verify our results.

The current study has important implications for the development of specific programs and interventions based on fostering and promoting the use of OR2-LMC technology among PD patients, and for the elimination of barriers and difficulties associated with treatment and relating to patients’ families. Our results may help professionals to understand people receiving novel technological treatment better, and to encourage the use of OR2-LMC therapies for UL rehabilitation in people with PD.

## 5. Conclusions

The OR2-LMC system, along with the serious games designed for it, represents a rehabilitation tool that could enhance certain UL outcomes in PD patients. However, elements of discordance, expansion, and confirmation between the qualitative and quantitative results of this study were identified. Further studies are necessary to corroborate and verify the effects of immersive VR technology combined with conventional therapies for the rehabilitation of patients with PD.

## Figures and Tables

**Figure 1 sensors-20-02168-f001:**
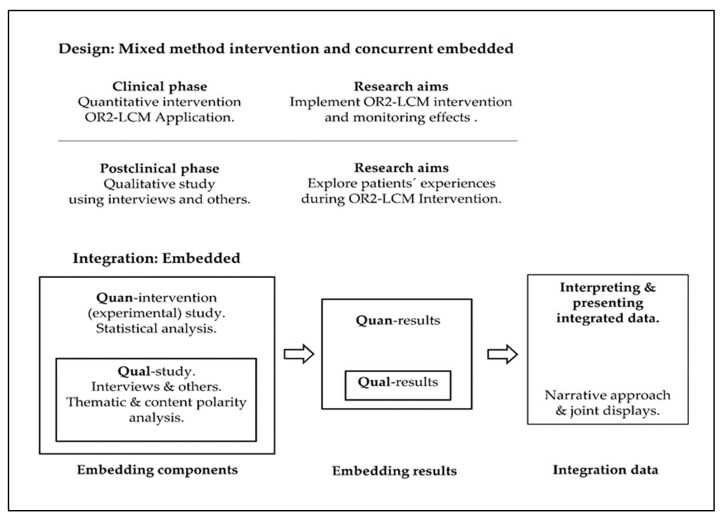
Mixed methods design and embedded integration. Quan, quantitative; Qual, qualitative.

**Figure 2 sensors-20-02168-f002:**
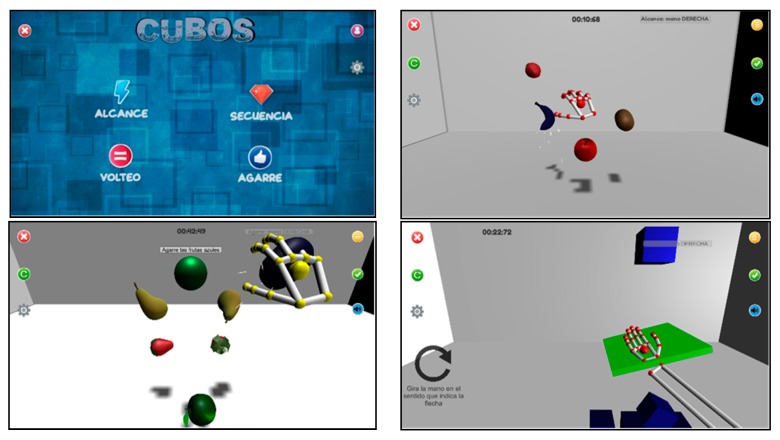
The designed video games: the reach game, the sequence game, the grab game, and the flip game). Note: “Alcance” = Reach; “Agarre” = Grip; “Secuencia” = Sequence; “Cubos” = Cubes; “Volteo” = Flip.

**Figure 3 sensors-20-02168-f003:**
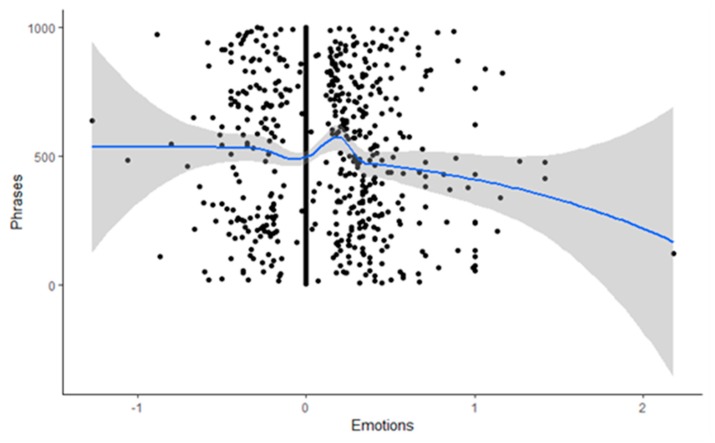
Polarity scatter plot.

**Figure 4 sensors-20-02168-f004:**
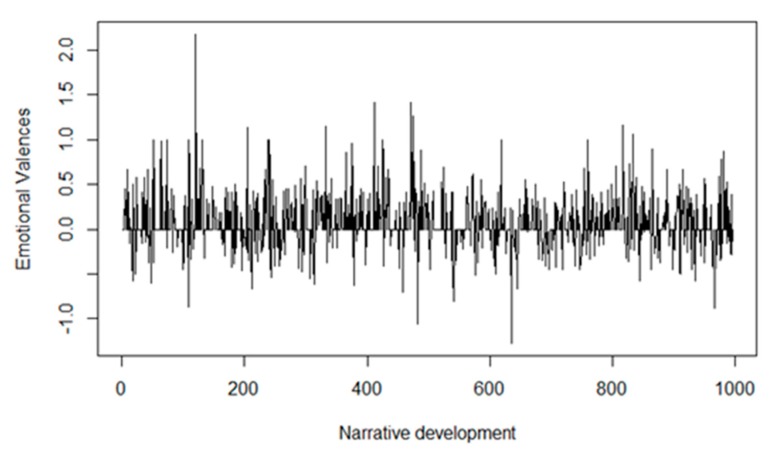
The emotional valences during narrative development. The horizontal axis represents the number of sentences of the six interviews, beginning with sentence 1 of interview 1 and ending with sentence 1000 of interview 6; the vertical axis represents the emotional valences, which have unlimited negative or positive values.

**Figure 5 sensors-20-02168-f005:**
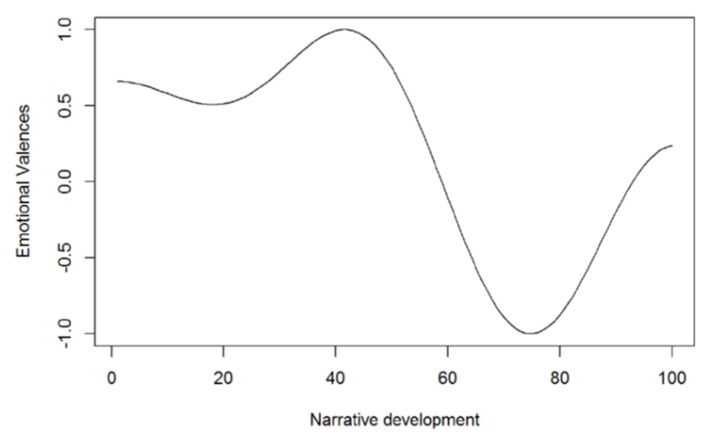
Evolution of polarity. The horizontal axis represents the normalized narrative development, beginning with interview 1 and ending with interview 6, and the clustering of phrases into groups of 10; the vertical axis represents the emotional valences, which have unlimited negative or positive values.

**Table 1 sensors-20-02168-t001:** Mixed methods intervention study summary.

	Component	Sampling	Participants	Intervention	Data Collection	Analysis
**Main study**	OR2-LMC novel technology intervention (non-randomized)	Non-probabilistic sampling of non-consecutive cases	Patients with PD (diagnostic criteria of the Brain Bank of the United Kingdom; stages II, III, and IV of the Hoehn and Yahr scale; >60% Schwab and England functionality scale	Oculus Rift 2 with the leap motion controller intervention using virtual reality	A Jamar^®^ hydraulic hand dynamometer (grip strength), the block and box test (unilateral gross manual dexterity), the Purdue pegboard test (coordination, speed of movement, and fine motor dexterity), the action research arm test (upper extremity performance), and the client satisfaction questionnaire (the satisfaction of health service users)	The statistical analysis was performed using the SPSS statistical software system. The Shapiro-Wilk’s test and the Kolmogorov-Smirnov test were used to screen all data for normality of distribution. Additionally, the Wilcoxon test for related samples was used to compare variables.
**Embedded study**	A qualitative case study	Purposeful sampling and information power criteria	The same ncludi ncluding at main study	Non-applicable	Semi-structured interviews based on a question guide and researcher field notes	Thematic inductive analysis and content analysis of free text using The Bing Dictionary and SODictionariesV1.11Spa2 to obtain a description of the polarity (acceptance or rejection).

QUAN, quantitative; QUAL, qualitative; OR2-LMC, Oculus Rift 2 with leap motion controller; PD, Parkinson’s disease.

**Table 2 sensors-20-02168-t002:** Patient features.

Patients (n)	Age (Years) Mean (± Standard Deviation)	Gender	Hoenhn and Yahr	More Affected Side	Schwab and England Score (%) Mean (±Standard Deviation)
6 patients	74.50 (±4.72)	5 Male	II (2)	2 Right	71.66 (±9.83)
		1 Female	III (4)	4 Left	

**Table 3 sensors-20-02168-t003:** Comparison of outcome scores between the pre- and post-treatments.

Variable		Median (Interquartile Range)		*p*-Value		Cohen’s d
Jamar	More affected	Pre	26.85 (10.67)	Pre-Post	0.028 *	0.53
		Post	31.16 (9.25)	Pre-Follow-Up	0.093	0.32
		Follow-up	28.00 (10.00)	Post-Follow-Up	0.062	0.21
	Less affected	Pre	23.13 (13.33)	Pre-Post	0.028 *	0.54
		Post	30.66 (14.33)	Pre-Follow-Up	0.028 *	0.15
		Follow-up	29.83 (15.00)	Post-Follow-Up	0.136	0.47
BBT	More affected	Pre	42.50 (22.00)	Pre-Post	0.039 *	0. 57
		Post	46.00 (16.50)	Pre-Follow-Up	0.916	0.10
		Follow-up	44.50 (11.50)	Post-Follow-Up	0.058	0.21
	Less affected	Pre	50.00 (9.25)	Pre-Post	0.343	0.18
		Post	49.00 (14.25)	Pre-Follow-Up	0.684	0.16
		Follow-up	48.00 (15.25)	Post-Follow-Up	0.715	0.25
PPT	More affected	Pre	7.83 (4.92)	Pre-Post	0.027 *	0.57
		Post	8.66 (4.50)	Pre-Follow-Up	0.073	0.20
		Follow-up	8.16 (4.75)	Post-Follow-Up	0.109	0.26
	Less affected	Pre	8.66 (2.50)	Pre-Post	0.028 *	0.54
		Post	9.83 (2.75)	Pre-Follow-Up	0.400	0.37
		Follow-up	8.16 (5.66)	Post-Follow-Up	0.686	0.11
PPT both hands		Pre	11.00 (5.17)	Pre-Post	0.168	0.15
		Post	11.33 (6.00)	Pre-Follow-Up	0.026 *	0.11
		Follow-up	11.99 (5.17)	Post-Follow-Up	0.715	0.19
PPT assemblies		Pre	12.16 (8.92)	Pre-Post	0.028 *	0.57
		Post	13.83 (10.33)	Pre-Follow-Up	0.600	0.26
		Follow-up	12.49 (5.33)	Post-Follow-Up	0.416	0.14
ARAT total score	More affected	Pre	52.50 (8.25)	Pre-Post	0.180	0.15
		Post	53.50 (4.50)	Pre-Follow-Up	0.276	0
		Follow-up	53.50 (5.25)	Post-Follow-Up	0.679	0.14
	Less affected	Pre	52.50 (6.25)	Pre-Post	0.596	0.10
		Post	53.00 (2.50)	Pre-Follow-Up	0.891	0.21
		Follow-up	54.00 (6.25)	Post-Follow-Up	0.914	0.24

ARAT, action research arm test; BBT, box and block test; PPT, Purdue pegboard test. Data are expressed as median and interquartile ranges. Note: * *p* value < 0.05 using the Wilcoxon test for related sample. Cohen’s D was used to estimate effect size for comparisons.

**Table 4 sensors-20-02168-t004:** The client satisfaction questionnaire (CSQ-8).

Variable	Patients Punctuation
1. Quality of service	3.66 (0.51)
2. Type of service	3.33 (0.51)
3. Needs met	3.66 (0.51)
4. Recommend to a friend	3.66 (0.51)
5. Amount of help	4 (0)
6. Deal with problems	3.33 (0.51)
7. Overall satisfaction	4 (0)
8. Come back	3.66 (0.51)
Total Score	3.66 (0.18)

Data are expressed as the mean and standard deviation.

**Table 5 sensors-20-02168-t005:** Combined display of the quantitative and qualitative findings.

Outcomes	Quantitative Findings	Qualitative Findings	Observations
Jamar: measures grip strength	Significant improvements on the Jamar post-treatment and at follow-up	There was no narrative regarding strength improvement	Patients initially reported no perceived short-term benefits from the intervention, although later on at home, patients did report improvements in their activities of daily living. Patients believed the intervention was aimed at achieving mental and cognitive (concentration, reaction rate, memory, etc.) improvements. Patients reported their experiences using the OR2-LMC system, describing it as a process that began with nervousness, fear, and surprise with the virtual environment. Through adaptation to the virtual world and confronting these new challenges, they eventually gained control of the tests and treatment. This process resulted in satisfaction (through overcoming the challenges and limitations), frustration (needing to train more), or boredom (monotony of the games).
BBT: measures unilateral gross manual dexterity	Significant and positive results for the affected side post-treatment	Patients reported improvement in activities such as handling dishes and cutlery	
PPT: measures coordination, speed of movement, and fine motor dexterity	Significant improvements for both hands post-treatment, both hands at follow-up, and assembly capacity post-treatment	Patients reported improved fine movements in activities such as accepting a purchase ticket. Coordination and the ability to overcome obstacles were also improved, although no improvements in the speed of movements were reported	
ARAT: measures upper limb performance	No significant results	Patients reported improved movements when handling dishes, objects, and reaction speeds	
CSQ-8: measures patient satisfaction	High degree of satisfaction; patients obtained a mean of 3.66 (0.18) points out of the maximum of 4	Polarity results showed a general acceptance of OR2-LMC therapy, although some patients pointed out the necessity of more clearly explaining the treatment and realistic expectations for its use; some patients were unaware of the objectives of the study or expected greater effects from therapy	

ARAT, action research arm test; BBT, box and block test; CSQ-8, client satisfaction questionnaire; PPT, Purdue pegboard test.; Jamar, JAMAR^®^ hydraulic hand dynamometer.

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
