# Peer review of "The Impact of a Novel Immersive Virtual Reality Technology Associated with Serious Games in Parkinson’s Disease Patients on Upper Limb Rehabilitation: A Mixed Methods Intervention Study"

_sensors, 2020, doi:10.3390/s20082168_

Round 1

Reviewer 1 Report

This manuscript studies the effects of the immersive virtual reality technology associated with serious games (Oculus Rift 2 plus Leap Motion Controller, OR2-LMC) on upper limb outcomes (muscle strength, coordination, speed of movements, fine and gross dexterity) in patients with Parkinson´s disease.  While the intervention showed positive results on the upper limb, there were elements of discordance.  Please also address the following comments before consideration for publication.

1. The authors may want to benchmark the performance results from this study against those from literature reports to directly demonstrate the novelty of this study.

2. Certain figure plots (Figs. 3-5) might be combined for a concise presentation.

3. The sample size of 6 seems to be relatively small.  Is it statistically sufficient to support the conclusion?

4. Please elaborate on the results (e.g., Figs. 6-7) and provide the physical insights into the results.

5. Please elaborate on the discussion of the discordance.

6. The manuscript would benefit from professional editing to help correct grammatical issues and typos.

7. Please remove the editing tracks.

Author Response

RESPONSE LETTER

Sensors

Manuscript ID: sensors-757743

Title: The impact of a novel immersive virtual reality technology associated with serious games in Parkinson's Disease patients for upper limb rehabilitation: a mixed method intervention study.

Reviewers' comments:

Reviewer 1

This manuscript studies the effects of the immersive virtual reality technology associated with serious games (Oculus Rift 2 plus Leap Motion Controller, OR2-LMC) on upper limb outcomes (muscle strength, coordination, speed of movements, fine and gross dexterity) in patients with Parkinson´s disease.  While the intervention showed positive results on the upper limb, there were elements of discordance.  Please also address the following comments before consideration for publication.

Response: Thank you very much for your comments. We really appreciate them. All the suggestions have been incorporated to this revised version of the paper.

The authors may want to benchmark the performance results from this study against those from literature reports to directly demonstrate the novelty of this study.

Response: Discussion section has been re-written.

Certain figure plots (Figs. 3-5) might be combined for a concise presentation.

Response: Figures 3 to 5 have unified.

 The sample size of 6 seems to be relatively small. Is it statistically sufficient to support the conclusion?

Response: Data analysis have been improved and completed. Mean and the standard deviation of parameters were used to calculate the effect size for the comparisons using the Cohen’s d statistic. Mean differences of 0.2, 0.5, and 0.8 standard deviations are considered ‘small’, ‘medium’, and ‘large’ effect sizes, respectively. All this information has been incorporated to Results and Discussion section. Despite to a small sample of patients, there are variables with statistical significance that show a medium effect size.

 Please elaborate on the results (e.g., Figs. 6-7) and provide the physical insights into the results.

Response: I agree with reviewer. We have included more information about the analysis process in the method section. In addition, more information is included in the results, the results are described more accurately step by step, and one table is added and two figures are replaced in the supplementary material.

We included new table S4.

We included new figure S1 and figure S2.

 Please elaborate on the discussion of the discordance.

Response: We agree with the reviewer. Discussion section has been re-written. Also, regarding discordance of results we have included more information at discussion section.

We have included new references:

61.Tsai, L.L.Y.; McNamara, R.J.; Dennis, S.M.; Moddel, C.; Alison, J.A.; McKenzie, D.K.; McKeough, Z.J. Satisfaction and Experience With a Supervised Home-Based Real-Time Videoconferencing Telerehabilitation Exercise Program in People with Chronic Obstructive Pulmonary Disease (COPD). Int J Telerehabil, 2016, 8(2),27-38. doi: 10.5195/ijt.2016.6213.

62.Coventry, P.; Bower, P.; Blakemore, A.; Baker, E.; Hann, M.; Li, J.; Paisley, A.; Gibson, M. Satisfaction with a digitally-enabled telephone health coaching intervention for people with non-diabetic hyperglycaemia. NPJ Digit Med, 2019, 2, 5. doi: 10.1038/s41746-019-0080-6.

63.Kraepelien, M.; Svanborg, C.; Lallerstedt, L.; Sennerstam, V.; Lindefors, N.; Kaldo, V. Individually tailored internet treatment in routine care: A feasibility study. Internet Interv, 2019, 18, 100263. doi: 10.1016/j.invent.2019.100263.

64.Kringle, E.A.; Setiawan, I.M.A.; Golias, K.; Parmanto, B.; Skidmore, E.R. Feasibility of an iterative rehabilitation intervention for stroke delivered remotely using mobile health technology. Disabil Rehabil Assist Technol, 2019, 1-9. doi: 10.1080/17483107.2019.1629113.

65.Mawhinney, G.; Thakar, C.; Williamson, V.; Rothenfluh, D.A.; Reynolds, J. Oxford Video Informed Consent Tool (OxVIC): a pilot study of informed video consent in spinal surgery and preoperative patient satisfaction. BMJ Open, 2019, 9(7), e027712. doi: 10.1136/bmjopen-2018-027712.

 The manuscript would benefit from professional editing to help correct grammatical issues and typos.

Response: We have followed the reviewer's recommendations. The manuscript has been revised by an English translation company. Translation certificate included.

Please remove the editing tracks.

Response: We have followed the reviewer's recommendations. Editing tracks have been removed.

 We strongly believe that the comments and recommendations by the editor and the reviewer have improved the quality of the text. As a result, we believe this has improved the clarity and presentation of the manuscript. We hope that the current version is now feasible for publication in the Journal.

Many thanks for your recommendations.

Kind regards

Sincerely

The authors

Reviewer 2 Report

Authors applied a mixed method intervention to evaluate the effects of a novel immersive virtual reality with serious game on upper limb rehabilitation related to PD patients. Manuscript is well written and well strctured.

My only concerns is related to the low number of subjects. Did authors evaluate the effect size of the study? The power of statistical analysis should be added and, likely, discussed in the limitations of the study.

In addition, how have the three rater assessments been treated? It is not clear if three raters were used for each patient. If yes, I would like to suggest adding an inter-rater analysis (for example through ICC) to evaluate the repeatability of the measurements.

Finally, I suggest to add some numeric results in the abstract.

Author Response

RESPONSE LETTER

Sensors

Manuscript ID: sensors-757743

Title: The impact of a novel immersive virtual reality technology associated with serious games in Parkinson's Disease patients for upper limb rehabilitation: a mixed method intervention study.

Reviewers' comments:

Reviewer 2

 Authors applied a mixed method intervention to evaluate the effects of a novel immersive virtual reality with serious game on upper limb rehabilitation related to PD patients. Manuscript is well written and well structured.

Response: Thank you very much for your comments. We really appreciate them. All the suggestions have been incorporated to this revised version of the paper.

My only concerns is related to the low number of subjects. Did authors evaluate the effect size of the study? The power of statistical analysis should be added and, likely, discussed in the limitations of the study.

Response: Data analysis have been improved and completed. Mean and the standard deviation of parameters were used to calculate the effect size for the comparisons using the Cohen’s d statistic. Mean differences of 0.2, 0.5, and 0.8 standard deviations are considered ‘small’, ‘medium’, and ‘large’ effect sizes, respectively. All this information has been incorporated to Results and Discussion section. Despite to a small sample of patients, there are variables with statistical significance that show a medium effect size.

In addition, how have the three rater assessments been treated? It is not clear if three raters were used for each patient. If yes, I would like to suggest adding an inter-rater analysis (for example through ICC) to evaluate the repeatability of the measurements.

Response: This section has been clarified. All outcome measures were conducted by three expert raters (rater 1 conducted Jamar and BBT assessment; rater 2 conducted PPT measures; rater 3 conducted ARAT and CSQ-8 evaluations) trained in all assessments and blinded to the interventions. Three evaluation time periods were determined: prior to any intervention, in a post-treatment period (post-six weeks period of treatment) and in a follow-up session (one month without rehabilitation treatment).

Finally, I suggest to add some numeric results in the abstract.

Response: This section has been reviewed following your recommendations. Thank you very much.

We strongly believe that the comments and recommendations by the editor and the reviewer have improved the quality of the text. As a result, we believe this has improved the clarity and presentation of the manuscript. We hope that the current version is now feasible for publication in the Journal.

Many thanks for your recommendations.

Kind regards

Sincerely

The authors

Round 2

Reviewer 1 Report

The authors have addressed the previous comments.

Reviewer 2 Report

Authors properly answered to my comments